# ATTRIBUTING MODEL BEHAVIOR: THE PREDOMINANT INFLUENCE OF DATASET COMPLEXITY OVER HYPERPARAMETERS IN CLASSIFICATION

## ABSTRACT

Understanding the drivers of machine learning performance is essential for optimizing model accuracy and robustness. While significant attention has been given to hyperparameter tuning and data preprocessing, the impact of intrinsic data complexity (e.g., class overlap, feature overlap, dimensionality, etc) remains less explored. This study investigates the comparative influence of data complexity and hyperparameter configurations on the performance of classification algorithms, specifically Random Forests (RF), Support Vector Machines (SVM), Decision Tree (DT), Adaptive Boosting (AB) and Multi-layer Perceptron (MLP). Using 270 diverse OpenML datasets and 304 hyperparameter configurations, we employ functional analysis of variance (fANOVA) and Ordinary Least Squares (OLS) regression to quantify the relative importance and effect sizes of hyperparameters and complexity meta-features. Our results reveal that data complexity exerts a more substantial influence on both bias and variance components than hyperparameter tuning, underscoring the importance of addressing intrinsic dataset challenges. These findings suggest that efforts to mitigate data complexity factors, such as class overlap or imbalance, may yield greater performance improvements than extensive hyperparameter optimization. This study provides actionable insights for machine learning practitioners and highlights the need for further research into the interplay between dataset properties and algorithmic performance.

## 1 INTRODUCTION

The performance of machine learning models is influenced by various factors, including the quality of the data, the choice of algorithms, and the optimization of hyperparameters. While considerable research has focused on techniques for hyperparameter tuning to improve model performance, less attention has been given to the inherent properties of datasets—referred to as data complexity meta-features—that fundamentally shape a model's ability to learn effectively.

Hyperparameters govern the complexity and architecture of machine learning models, necessitating precise selection prior to model training. The process of identifying the optimal hyperparameters to maximize the performance of a machine learning algorithm on a given dataset is known as hyperparameter tuning. This involves exploring a range of hyperparameter settings using various optimization techniques such as Grid search (Lerman, 1980), Random search (Bergstra & Bengio, 2012), and Bayesian optimization (Mockus, 1982). However, this optimization can be computationally intensive, particularly for complex models like deep neural networks, which are characterized by numerous hyperparameters. Understanding the influence of hyperparameters has become increasingly important, as identifying the critical hyperparameters can narrow the optimization space, thereby improving efficiency. Consequently, several studies (Biedenkapp et al., 2017; Huang & Boutros, 2016; Hutter et al., 2014; Jin, 2022; van Rijn & Hutter, 2018; Taylor et al., 2021; Trithipkaiwanpon & Taetragool, 2021) have been dedicated to quantifying and ranking the influence of hyperparameters on classification performance. Notably, in (Hutter et al., 2014), the authors introduced a functional analysis of variance (fANOVA) framework to quantify the individual and pairwise contributions of hyperparameters. Their findings underscored that model performance could often be attributed to a

select few hyperparameters. Utilizing the fANOVA framework, van Rijn & Hutter (2018) examined the most influential hyperparameters of various algorithms, including Random Forests (RF), Support Vector Machines (SVM), and Adaptive Boosting (AB), across 100 datasets, revealing consistent key hyperparameters across these datasets.

It is equally well established that the predictive performance of classifiers is influenced by the complexity of the given dataset. To elucidate this dependency, complexity meta-features have been proposed (Ho & Basu, 2002; Mollineda et al., 2005; Ho et al., 2006; Sotoca et al., 2006; Orriols-Puig et al., 2010) to describe classification task difficulty by measuring aspects such as feature overlap, class overlap, linear separability, and feature dimensionality. In Cano (2013), researchers explored the impact of complexity meta-features on binary classification accuracy using synthetic datasets to control data complexity. They determined that feature-based measures had the most significant impact on classification accuracy, followed by separability measures. Similarly, Östlund & Fahlman (2022) analyzed multiclass classification tasks using SVM, k-Nearest Neighbors (KNN), and Multilayer Perceptron (MLP) on both real-world and synthetic datasets, investigating the influence of complexity meta-features on classification accuracy and variability across data folds.

Recent studies (Farhangi, 2022; Gardner et al., 2023; Talaei Khoei et al., 2023; Lima et al., 2024) have also investigated the comparative influence of data preprocessing and hyperparameter tuning on model performance. The findings of these studies reveal that performing data preprocessing improved model performance more than hyperparameter tuning. However, these approaches primarily address external transformations rather than the underlying complexity of the data and as such the foundational impact of data complexity on model behavior remains poorly quantified. Addressing this gap is crucial, as understanding the interplay between data complexity and hyperparameter configurations can inform the design of more robust and efficient machine learning systems.

This study builds upon prior research by systematically investigating the relative importance of data complexity and hyperparameter configurations on classification performance. Specifically, we perform a bias-variance decomposition of classification error, providing a granular perspective on how these factors influence both bias and variance components. Using functional ANOVA (fANOVA) and Ordinary Least Squares (OLS) regression, we quantify the contributions of complexity meta-features and hyperparameters, offering a novel approach to dissecting performance drivers.

This paper is structured as follows: Section 2 provides foundational knowledge on bias-variance decomposition and fANOVA. Section 3 outlines our experimental approach. Section 4 presents and discusses the experimental results.Section 5 highlights the contribution and implications of our study. Finally, Section 6 concludes with a summary of key findings and suggests avenues for future research. Section 7 in the appendix discusses our causal analysis.

## 2 BACKGROUND

This section provides foundational knowledge on bias-variance decomposition, and fANOVA.

### 2.1 BIAS-VARIANCE DECOMPOSITION

In classification tasks, bias refers to errors due to oversimplified assumptions about the problem, while variance quantifies how much predictions vary across different training samples. Recent research has emphasized the importance of decomposing a learner's error into bias and variance components, offering valuable insights into predictive performance. This approach, originating from regression analysis, has been extended to classification (Geman et al., 1992) and further developed by Domingos (2000).

In this study, we adopt this unified bias-variance decomposition approach to help us understand the expected loss by breaking it down into three components: noise, bias, and variance. Noise represents the inherent unpredictability in the data, bias represents the error due to assumptions made by the model, and variance represents the error due to the variability in the training data. These components provide a clearer picture of the sources of error in the model's predictions. We employ this decomposition approach because it is currently the only method that provides the most intuitive and general formulation of bias-variance for classification problems. For a comprehensive mathematical description of bias-variance decomposition, please refer to (Domingos, 2000).

## 2.2 FUNCTIONAL ANALYSIS OF VARIANCE

Functional analysis of variance is a technique used to assess the individual contributions of various components within a function to its overall variability. In this context, these contributions are referred to as performance metrics. During the training of a classifier on a dataset, multiple factors influence the training process. Each factor has a specific domain of possible values, and the combination of these domains forms a configuration. The performance of the classifier, when trained and evaluated on the dataset, is represented by a real-valued metric.

fANOVA breaks down the total variability in the classifier's performance into parts attributed to different subsets of these factors. This decomposition helps in understanding how much each subset of factors contributes to the overall variability. By quantifying these contributions, fANOVA provides insights into which factors are most influential in affecting the performance. For detailed definitions and efficient computation methods, please refer to the work of Hutter et al. (2014). In this study, we utilize the fANOVA framework to quantify the contributions of each hyperparameter and complexity meta-feature to the bias and variance of the classifier.

## 3 METHODOLOGY

This section outlines the methodological framework adopted in our study. It details the data selection, preprocessing, classification algorithm selection, meta-feature extraction, and bias-variance decomposition phases as depicted in Figure 1. The entire process was designed to examine the impact of hyperparameters and complexity meta-features on the bias-variance in machine learning models.

### 3.1 DATA SELECTION AND PREPROCESSING

We sourced 270 binary classification datasets from the OpenML repository (Vanschoren et al., 2014). OpenML is a widely used platform for sharing machine learning datasets and experiment results. Our dataset selection followed the criteria established by Mantovani et al. (2019) to ensure varied complexity levels across the datasets. The selection criteria were as follows:

- Instances per dataset: Between 100 and 50,000.
- Features per dataset: Maximum of 1,500.
- No missing values in the datasets.
- Datasets must not be simplified, altered, or binarized versions of multi-class problems.
- Exclusion of datasets that are adaptations of regression tasks.

Given that the selected datasets had no missing values, the only preprocessing step required was encoding categorical features. We used scikit-learn's label encoding (Pedregosa et al., 2011) to assign integer values to categorical variables. Other encoding techniques like one-hot encoding were avoided to prevent dimensionality inflation, especially in datasets with many categories. No normalization or scaling techniques were applied, as these could inadvertently modify the intrinsic complexity of the datasets. After encoding, each dataset was split into training (70%) and test (30%) sets for subsequent processes.

### 3.2 ALGORITHM SELECTION AND HYPERPARAMETER CONFIGURATION

#### 3.2.1 ALGORITHM SELECTION

We employed two well-known supervised classification algorithms for this study: Random Forest (RF), a robust ensemble method that combines decision trees to improve predictive accuracy and reduce overfitting. Support Vector Machine (SVM), a classification algorithm that optimizes a decision boundary (hyperplane) to maximize class separation in the feature space. Decision Tree (DT), a tree-based model that splits data based on feature rules to classify it. Adaptive Boosting (AB), an ensemble method that combines multiple weak learners sequentially, focusing on fixing prior errors. Multi-layer Perceptron (MLP): a type of artificial neural network with one or more hidden layers, which learns complex patterns through backpropagation by adjusting weights based on errors

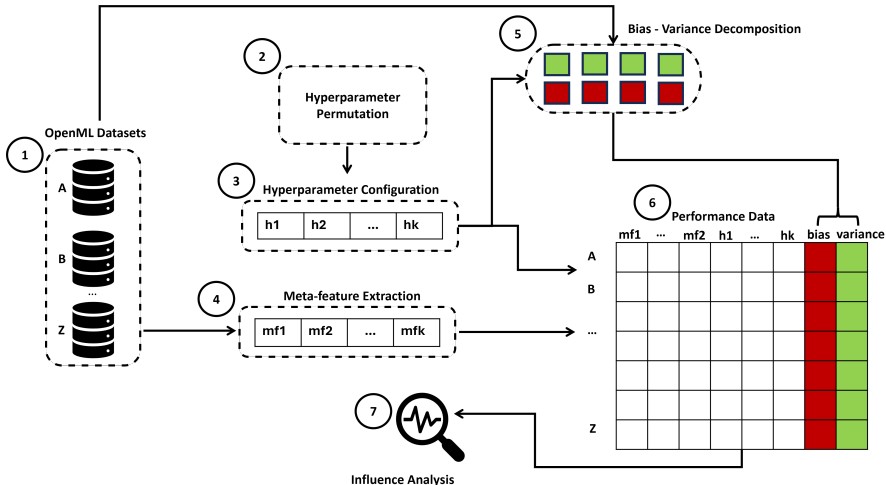

Figure 1: A framework for assessing the contributions of hyperparameters and complexity meta-features to bias and variance.

These models were chosen because they represent different algorithmic paradigms— DT as a tree-based method, RF and AB as a tree-based ensemble method, SVM as a non-tree-based method and MLP as a neural network method—allowing us to generalize the effect of hyperparameters and complexity meta-features across model types.

### 3.2.2 HYPERPARAMETER CONFIGURATION

To explore the influence of hyperparameter tuning, we defined a search space for each model based on prior research by van Rijn & Hutter (2018). The key hyperparameters for all the models were selected for permutation within this search space. The search space was designed to ensure broad coverage of hyperparameter settings while avoiding computational intractability. In total, we generated 304 distinct hyperparameter configurations for each model (Stages 2 & 3 in Figure 1). Detailed settings for the hyperparameters are provided in Table 1 in the appendix section.

### 3.3 COMPLEXITY META-FEATURE EXTRACTION

We extracted three key complexity meta-features that characterize the difficulty of classification tasks. These meta-features have been extensively studied in the literature (Ho & Basu, 2002; Lorena et al., 2012; Mollineda et al., 2005). The following meta-features were extracted using the **pymfe** package (Alcobaça et al., 2020):

- **Class Overlap (N1):** This complexity metric assesses whether the distributions of two class labels (class 0 and class 1) are separable. To evaluate class overlap, we employed the Fraction of Borderline Points method. This method constructs a minimum spanning tree connecting all data points, regardless of class labels, and counts the number of points connected to a different class via tree edges. A higher N1 value (ranging from 0 to just below 1) suggests significant overlap between the classes. .

- **Data Sparsity (T2):** This complexity metric evaluates the distribution density of data points within the input space. We measured data sparsity by comparing the ratio of data features to data samples. This method, referred to as the Average number of features per points, provides insight into the distribution density. Higher T2 values (bounded between 0 and $m$) indicate more sparse distributions, where $m$ represents the total number of features.

- **Class Imbalance (C2):** This complexity measure captures the degree of imbalance between the classes in the dataset. To measure class imbalance, we computed the ratio between the number of samples in the majority class and the minority class. Higher values of C2 (bounded between 0 and 1) indicate a greater imbalance between classes.

- **Feature Overlap (F1v):** This complexity measure evaluates the separability of two classes by identifying an optimal projection vector. Once data points are projected onto this vector, it assesses their separability using a directional Fisher criterion. F1v provides a bounded value within the interval (0, 1]. Lower F1v values indicate simpler classification tasks, as the classes are more easily distinguishable within the projected space.

- **Linear Separanility of Classes (L2):** The L2 complexity measure quantifies the performance of a linear support vector machine (SVM) by computing its error rate on the dataset. Higher L2 values (bounded between 0 and 1) suggest that the data is less amenable to linear separation, implying increased classification complexity. This metric highlights datasets where linear models struggle to achieve low error rates, often due to non-linear class boundaries or overlapping distributions.

- **Density:** This measure assesses data connectivity in a graph by normalizing the number of edges against the maximum possible. Lower Density values (bounded between 0 and 1) indicate dense, same-class clusters (simpler tasks), while sparse graphs with fewer edges or interclass proximity suggest higher complexity.

Each complexity meta-feature was calculated from the training set of each dataset (Stage 4 in Figure 1).

### 3.4 Bias-Variance Decomposition

To assess the impact of both hyperparameters and complexity meta-features on model performance, we conducted a bias-variance decomposition for each classification model (Stage 5 in Figure 1). Using the **bias_variance_decomp** function from the **MLxtend** library (Raschka, 2018), we trained each of the 304 hyperparameter configurations on each dataset and decomposed the classification error into bias and variance components using the 0-1 loss function. These components offer a more granular view of model performance beyond just classification accuracy.

### 3.5 Performance Data Generation and Influence Analysis

**Performance Data Generation:** The next phase involved consolidating the bias-variance results, hyperparameter configurations, and meta-feature values into a structured dataset referred to as performance data (Stage 6 in Figure 1). This dataset contained both the feature variables (hyperparameters and complexity meta-features) and target variables (bias and variance).

**fANOVA Analysis:** To quantify the relative influence of hyperparameters and complexity meta-features on bias and variance, we applied the fANOVA framework (Hutter et al., 2014) (Stage 7 in Figure 1). This technique enabled us to decompose the variation in bias and variance estimates and assign it to individual hyperparameters or complexity meta-features. We employed fANOVA because it is well-suited for high-dimensional parameter space exploration, allowing us to determine the relative importance of each factor. We normalized the independent variables to improve the interpretability of results.

**Linear Regression Analysis:** To complement fANOVA and quantify the magnitude and direction of each factor's effect, we applied Ordinary Least Squares (OLS) regression. We standardized the independent variables, which included hyperparameters and meta-features, and tested the regression coefficients for statistical significance using p-values. This approach enabled us to analyze and interpret the relationships between these factors and the resulting bias-variance outcomes.

**Causal Analysis:** Lastly, we conducted an analysis based on the Manipulation Theory of Causation (Section 7 in appendix) to establish a causal link between dataset complexity and model performance. This analysis confirmed that dataset complexity has a more consistent and pronounced effect on bias and variance than hyperparameters.

## 4 Experimental Results

### 4.1 fANOVA Analysis Results

This section presents the findings of our fANOVA analysis.

### 4.1.1 Bias Results

The fANOVA analysis (Figure 2) shows that complexity meta-features overwhelmingly dictate the bias in all the models considered in this study. These features collectively accounted for a substantial proportion of bias variability across all models: 74.0% for SVM, 65.6% for RF, 62.6% for DT, 52.0% for MLP, and 45.5% for AB. More specifically Figure 3 reveals that Class overlap (N1) emerged as the most significant contributor to bias in RF, DT, MLP, and AB, explaining 53.8%, 53.5%, 44.0%, and 28.2% of the variability, respectively. Notably, SVM deviated from this pattern, with class imbalance (C2) being the dominant factor, explaining 60.0% of its bias variability. This observation aligns with the known sensitivity of SVM to class distribution, favoring majority-class performance (Akbani et al., 2004). As the second most influential meta-feature, Density influenced SVM (12.2%) and AB (9.3%) bias significantly, while RF bias was more affected by feature overlapping (F1v; 6.0%). Across all models, hyperparameters played a relatively minor role, cumulatively explaining less than 5% of bias variability in each model. This reveals the limited direct influence of hyperparameter tuning on bias.

Interestingly, we deduced from our results that a substantial proportion of bias variability—ranging from 25.7% in SVM to 50.4% in AB—remains unexplained, suggesting potential interactions between data complexity features and hyperparameters that were not captured in this study. These unexplained variabilities warrant further exploration of higher-order and nonlinear interactions in future studies.

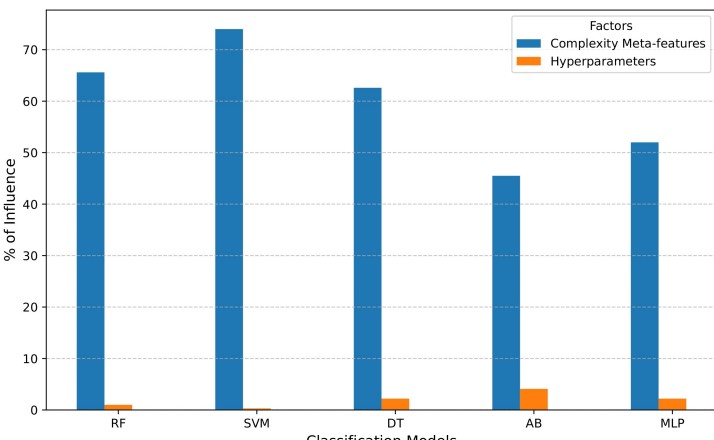

Figure 2: Comparison of the total influence of hyperparameters and complexity meta-features on bias.

### 4.1.2 Variance Results

In the analysis of variance, data complexity meta-features once again emerged as most influential (Figure 4), though with slightly reduced contributions compared to bias. These meta-features collectively explained 67.3% of variance variability in DT, 64.0% in RF, 59.0% in SVM, 40.4% in AB, and 26.1% in MLP. Again, N1 was the most impactful complexity meta-feature (Figure 5) for RF, DT, AB, and MLP, accounting for 56.8%, 49.6%, 27.4%, and 16.0% of their respective variance variability. SVM variance was most strongly influenced by C2 (46.1%), reinforcing its sensitivity to class imbalance. The role of hyperparameters in variance variability was slightly higher than in bias but remained secondary, with the highest contribution observed in MLP (9.4%) and AB (6.5%).

Also for variance, a significant portion of its variability remained unexplained, ranging from 31.1% in DT to 64.5% in MLP. This suggests that additional unmodeled interactions or external factors may significantly influence model variance.

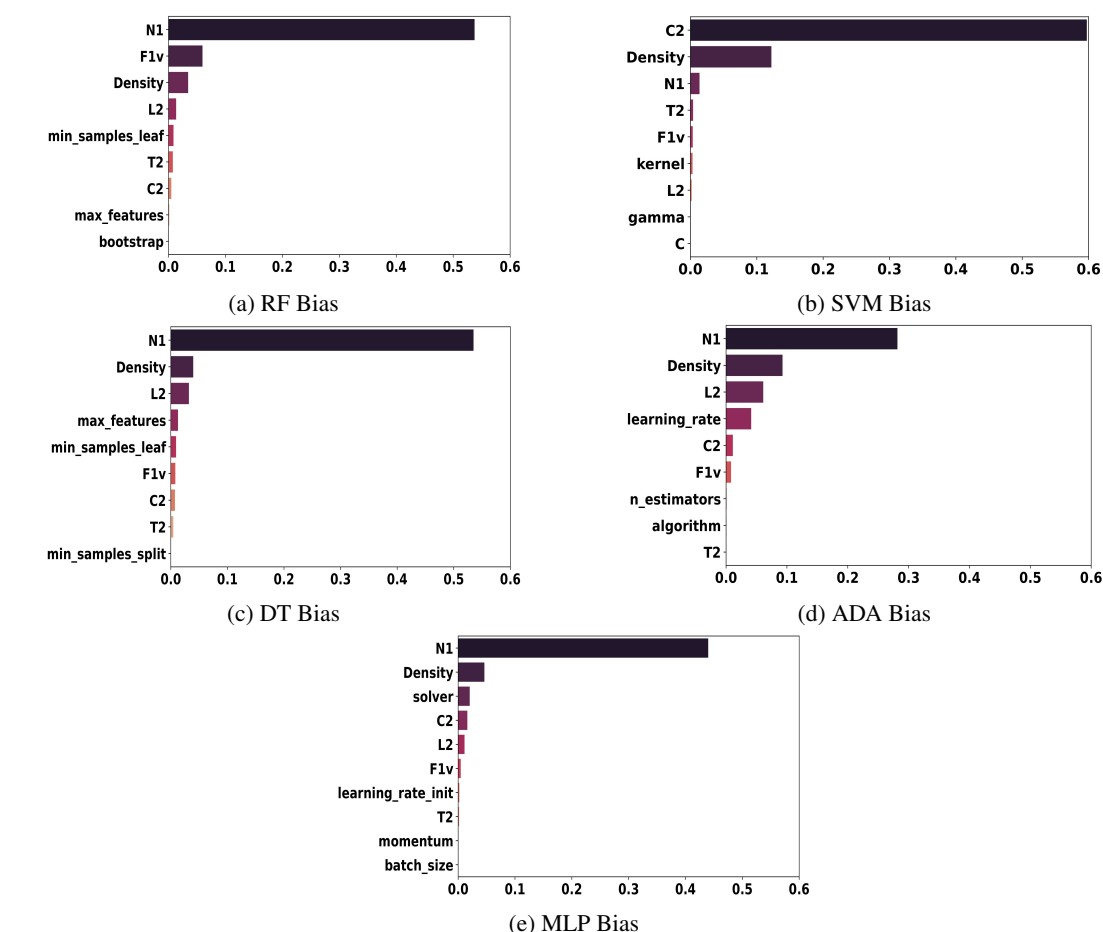

Figure 3: Comparison of the effects of hyperparameters and complexity meta-features on bias. The X-axis indicates the degree of contribution, while the Y-axis shows the influencing factors.

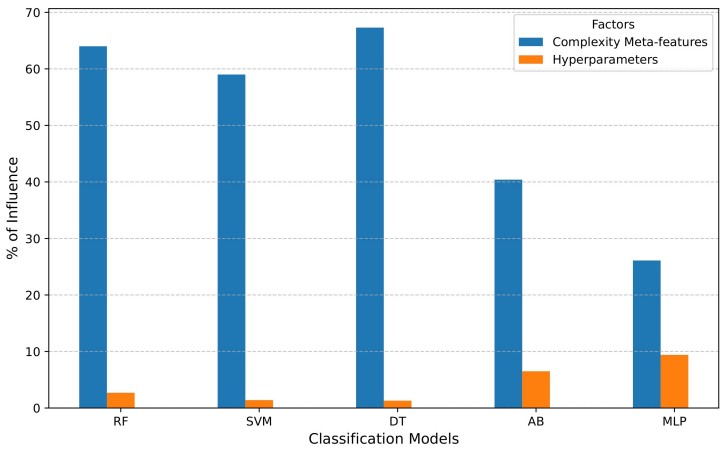

Figure 4: Comparison of the total influence of hyperparameters and complexity meta-features on variance.

## 4.2 REGRESSION ANALYSIS RESULTS

This section presents the findings of our Ordinary Least Squares (OLS) regression analysis (Table 2 to 11 in the appendix section). When interpreting the regression results, we primarily focus on the direction of the correlation for each predictor variable, as OLS measures only the linear relationships between the analyzed factors and performance. Hence, for rankings of influence, we rely on the results from fANOVA, as it captures complex, nonlinear dependencies between the factors and performance, which OLS cannot.

### 4.2.1 BIAS RESULT

**RF:** The linear regression analysis indicates that all predictor variables, including complexity meta-features and hyperparameters, significantly influence the bias of the RF model ($p < 0.001$). Among the complexity meta-features, N1 (coefficient = 0.0945), F1v (coefficient = 0.0408), and T2 (coefficient = 0.0184) exhibit the highest positive coefficients, suggesting their substantial role in increasing bias. In contrast, L2 (coefficient = 0.0144) and max_features (coefficient = 0.0062) show negative correlations, indicating that better class separation and increased max_features reduce RF bias. All RF hyperparameters contribute modestly but significantly, highlighting their subtle impact on bias adjustment.

**SVM:** For the SVM model, all predictor variables except the hyperparameter gamma significantly contribute to bias ($p < 0.001$). Complexity meta-features N1 (coefficient = 0.0155) and F1v (coefficient = 0.0211) have the highest positive coefficients, showing their strong influence on bias. Conversely, C2 (coefficient = 0.1139) demonstrates the largest negative impact, implying that imbalanced class distributions reduce SVM bias. Additionally, hyperparameters such as kernel and C exhibit small but statistically significant contributions.

**DT:** The analysis reveals that all predictor variables, except the complexity meta-feature L2 and the hyperparameter min_samples_split, significantly affect the bias of the DT model ($p < 0.001$). Complexity meta-features N1 (coefficient = 0.1054) and F1v (coefficient = 0.0399) show the highest positive contributions. The hyperparameters min_samples_leaf and max_features also display small but statistically significant impacts.

**AB:** For AB, all complexity meta-features significantly influence bias ($p < 0.001$). Among the complexity meta-features, N1 (coefficient=0.0792) and F1v (coefficient=0.0348) remained the highest positive coefficients. Among hyperparameters, only learning_rate shows a statistically significant contribution.

**MLP:** In the MLP model, all predictor variables significantly impact bias ($p < 0.001$). Among complexity meta-features, N1 (coefficient = 0.0896) has the highest positive contribution. All hyperparameters display small but statistically significant contributions.

### 4.2.2 VARIANCE RESULT

**RF:** The variance analysis for RF identifies N1 (coefficient = 0.0485) and F1v (coefficient = 0.0251) as the most influential positive predictors, mirroring their effect on bias. In contrast, the hyperparameters bootstrap (coefficient = 0.0085) and min_samples_leaf (coefficient = 0.0012) negatively correlate with variance, reflecting the bias-variance trade-off inherent in hyperparameter tuning.

**SVM:** All complexity meta-features significantly contribute to SVM variance. N1 (coefficient = 0.0174) and T2 (coefficient = 0.0205) exhibit the strongest positive impacts. Conversely, C2 (coefficient = 0.0738) shows the largest negative correlation, suggesting that imbalanced class distributions reduce SVM variance. Interestingly, hyperparameters gamma and C are statistically insignificant in their contribution to variance relative to complexity meta-features.

**DT:** All complexity meta-features significantly influence DT variance, with N1 (coefficient = 0.1054) having the highest positive coefficient. Similar to its effect on bias, the hyperparameter

min_samples_split shows a statistically insignificant impact relative to complexity meta-features, suggesting that it plays a negligible role in both bias and variance adjustment.

**AB:** For AB, all predictor variables significantly contribute to variance ($p < 0.001$). Complexity meta-feature N1 (coefficient = 0.0792) remains the most prominent positive factor. Hyperparameters learning_rate (coefficient = 0.0211) and n_estimators (coefficient = 0.0070) maintain their respective positive and negative relationships with variance. This finding suggests that tuning these hyperparameters may not result in a bias-variance trade-off, warranting further investigation.

**MLP:** In the MLP model, all predictor variables except F1v significantly affect variance. Among complexity meta-features, N1 (coefficient = 0.0443) retains its dominant positive influence. All hyperparameters display small but significant contributions.

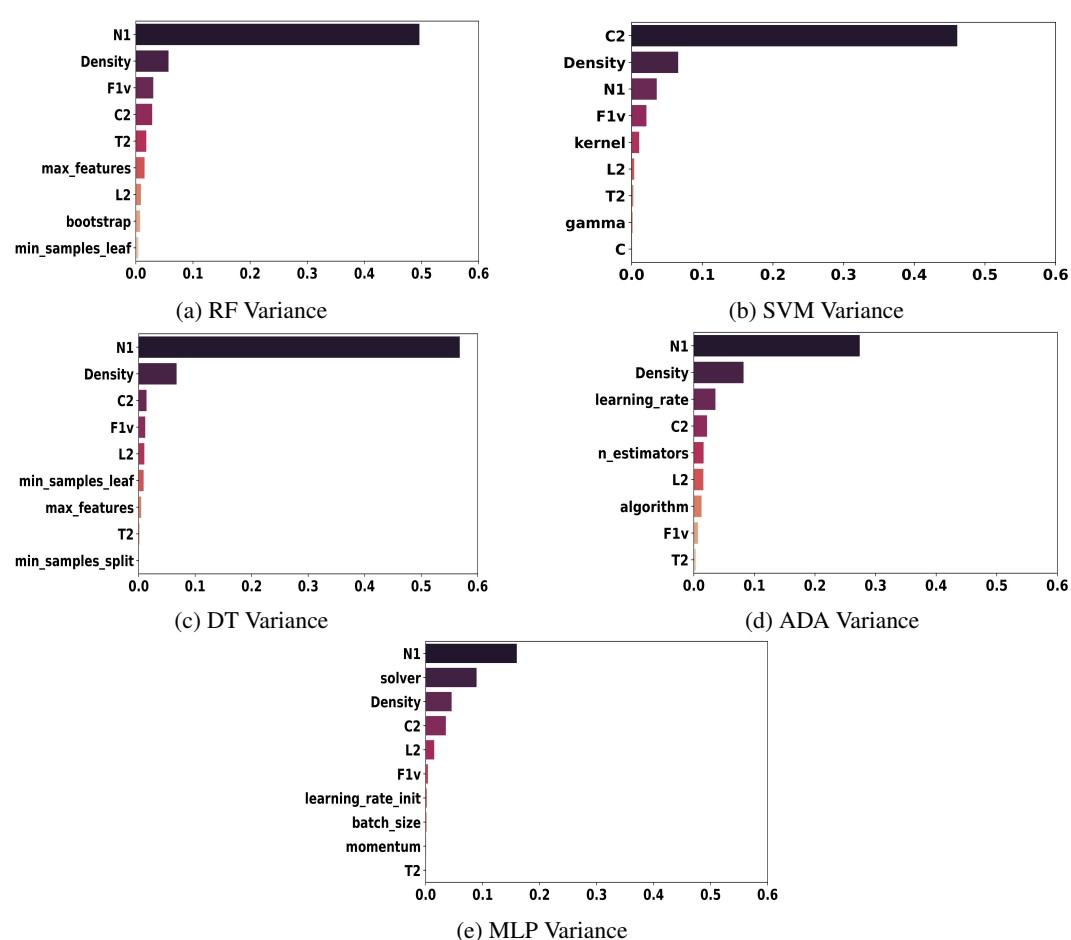

Figure 5: Comparison of the effects of hyperparameters and complexity meta-features on variance. The X-axis indicates the degree of contribution, while the Y-axis shows the influencing factors.

### 4.3 SUMMARY OF RESULTS

The findings emphasize that dataset complexity—particularly class overlap (N1) for tree-based models like Random Forest (RF) and Decision Trees (DT), and class imbalance for SVMs—plays a pivotal role in influencing bias and variance in classification models. Specifically, the significant impact of class overlap on RF and DT corroborates recent findings by Kim & Jung (2023), which identified improved performance in these models when resampling techniques such as SMOTE-TomekLinks and SMOTE-PSO were applied to datasets with high complexity (e.g., class overlap and imbalance). While that study reported performance improvements, it did not provide an explanation for the un-

derlying reasons. Our results fill this gap, demonstrating that N1 complexity strongly governs the bias-variance dynamics in tree-based models.

While hyperparameter tuning is essential, its impact diminishes for datasets with high complexity, where intrinsic characteristics exert a far greater influence. The unexplained variability in bias and variance underscores the need to explore higher-order interactions between complexity features and hyperparameters for a deeper understanding of model behavior.

Although uniform hyperparameter sampling may not capture narrow optimal ranges, the results robustly illustrate the dominance of dataset complexity over hyperparameters in determining performance variability.

## 5 IMPLICATIONS OF RESULTS

### 5.1 THEORETICAL IMPLICATIONS

**Importance of Data Complexity:** These findings emphasize the critical role of data complexity in classification performance. Beyond optimizing models, addressing intrinsic dataset challenges—such as class overlap—is essential to enhancing robustness and overall model effectiveness.

**Bias-Variance Tradeoff Insights:** The decomposition of classification error into bias and variance components revealed that complexity meta-features play a pivotal role in shaping these two elements. For example, data complexity features like N1 consistently demonstrated a positive correlation with both bias and variance, indicating that addressing this factor can simultaneously improve both aspects.

Another interesting insight was that for some models, the hyperparameters maintained the same correlation with both bias and variance. This suggests that tuning these hyperparameters could improve both bias and variance simultaneously, rather than leading to the typical bias-variance tradeoff often associated with hyperparameter tuning.

### 5.2 PRACTICAL IMPLICATIONS

**Data Preprocessing:** Practitioners should prioritize identifying and mitigating dataset complexities, such as class overlap, early in the development process. Focusing on these issues during data preprocessing can significantly enhance performance, often more effectively than hyperparameter optimization.

**Model Selection and Design:** For datasets with substantial complexity, selecting models inherently robust to class overlap, can offer more reliable reductions in bias and variance than relying solely on hyperparameter adjustments.

## 6 CONCLUSION

In this paper, we explored the relative impact of hyperparameters and dataset complexity meta-features on the bias-variance dynamics of machine learning models, using RF, SVM, DT, AB and MLP classifiers as case studies. Leveraging the fANOVA framework, we conducted an in-depth analysis to identify the primary drivers behind bias and variance, a research direction that has been largely underexplored. Our analysis showed that complexity meta-features—such as class overlap—have a consistently stronger impact on model behavior than hyperparameter settings. By highlighting the dominant role of dataset complexity in determining bias and variance, these findings emphasize the importance of adopting data-centric approaches to improving model performance.

Future research should extend this analysis to other domains and tasks to further validate these findings. Also, further work is needed to develop comprehensive meta-features that capture nuanced dataset properties beyond those studied here.

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

# 7 APPENDIX

## 7.1 CAUSAL ANALYSIS

To further validate the findings from our fANOVA analysis and reinforce the assertion that reducing dataset complexity has a greater causal impact on model performance than hyperparameter tuning, we conducted a causal analysis. Recognizing that *correlation does not imply causation*, we anchored this analysis in the Manipulation Theory of Causation, which posits that variable **X** is a direct cause of variable **Y** if an intervention on **X** results in a change in **Y**, while other factors remain constant (Galatis, 2018). This approach allows us to disentangle the effects of different interventions and attribute changes in model behavior to specific causes. It is important to note that we apply this theory under the assumption of causal sufficiency, meaning there are no hidden confounding factors influencing the observed relationships.

In this analysis, we aim to determine whether interventions on complexity meta-features—particularly the N1 complexity meta-feature (class overlap)—have a more profound and consistent impact on bias and variance than tuning the min_samples_leaf hyperparameter in an RF model. Our hypothesis is twofold:

- **H1:** N1 complexity exerts a stronger causal effect on both bias and variance than the min_samples_leaf hyperparameter.

- **H2:** Interventions to reduce N1 complexity will lead to greater improvements in overall model performance, compared to hyperparameter tuning.

This analysis seeks not only to verify causal relationships but also to attribute changes in model behavior to these interventions, shedding light on the role of data complexity in model generalization.

## 7.2 DATA GENERATION AND MODEL SETUP PROCESS

To evaluate this hypothesis, we utilized the **make_classification** function from scikit-learn (Pedregosa et al., 2011) to generate synthetic data that simulates the complexity characteristics of the German Credit Dataset (Hofmann, 1994). This synthetic dataset allowed us to control and manipulate class overlap to systematically observe the effect of the N1 complexity meta-feature on model bias and variance. The dataset was generated with the following specifications:

- Number of features: 20.

- Number of informative features: 10.

- Number of classes: 2.

- Class overlap: Configured to reflect a moderate N1 complexity value of 0.54.

- The class distribution was kept moderately imbalanced to retain a real-world characteristic, with a ratio of 60:40 for the two classes.

We employed an RF model as our base classifier for this analysis. The RF was initialized with default hyperparameter settings, except for the min_samples_leaf hyperparameter, which was intentionally misconfigured to a suboptimal value of 10 to simulate underfitting. This setup allows us to observe how hyperparameter tuning and dataset complexity reduction impact bias and variance under high-bias conditions.

## 7.3 HYPERPARAMETER AND N1 COMPLEXITY TUNING PROCESS

To assess the impact of hyperparameter tuning on model performance, we focused on optimizing the min_samples_leaf hyperparameter. We systematically varied the value of min_samples_leaf from 10 to 1. The goal was to observe the bias and variance behavior at different configurations of this parameter.

To evaluate the effect of reducing dataset complexity, we focused on decreasing the N1 complexity meta-feature by manipulating class overlap in the synthetic dataset. This was achieved by progressively separating the classes in feature space, thereby reducing the overlap between them. The

resulting changes in N1 allowed us to observe how decreasing class overlap influences bias and variance while keeping the hyperparameter min_samples_leaf fixed at its suboptimal value.

## 7.4 RESULTS

Our causal analysis results support both hypotheses (H1 and H2) and are consistent with the fANOVA findings, showing that dataset complexity has a significantly greater impact on bias and variance compared to hyperparameters.

**Effect of min_samples_leaf Tuning**: Figure 6a illustrates the impact of tuning min_samples_leaf on model performance. When the RF model was trained with the misconfigured value of min_samples_leaf, the model exhibited underfitting, characterized by high bias and low variance. Upon tuning min_samples_leaf to its optimal value of 1 (van Rijn & Hutter, 2018), we observed a marginal reduction in bias but a corresponding increase in variance. This confirms that while min_samples_leaf impacts bias, its effect on variance is limited, and the trade-off between bias and variance is consistent with typical hyperparameter optimization behavior.

**Effect of N1 Complexity Reduction:** In contrast, Figure 6b shows the effect of reducing N1 complexity by decreasing class overlap. We observed a simultaneous reduction in both bias and variance, despite the misconfigured min_samples_leaf hyperparameter. This result highlights that reducing class overlap and, by extension, dataset complexity, exert a stronger causal influence on model performance than hyperparameter tuning.

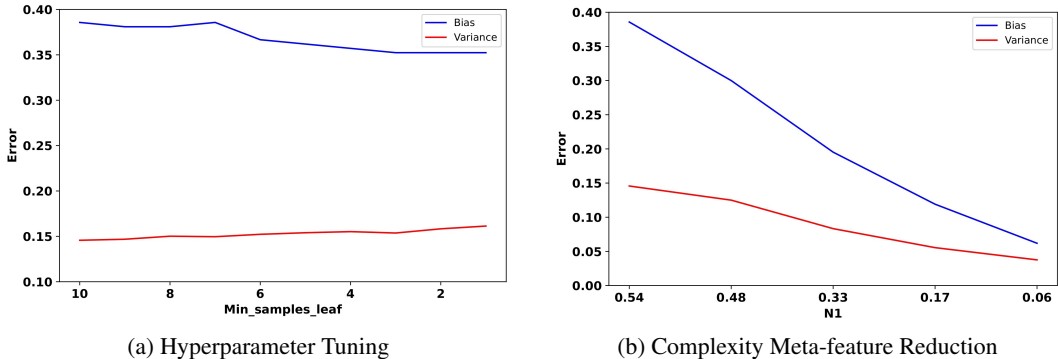

(a) Hyperparameter Tuning        (b) Complexity Meta-feature Reduction

Figure 6: Comparison of the impact of tuning the min_samples_leaf hyperparameter and the N1 complexity meta-features on bias and variance in RF models.

## 7.5 DISCUSSION

The causal analysis conducted in this study underscores the critical importance of dataset complexity, particularly class overlap, in shaping RF model behavior and performance. By isolating the effects of reducing the N1 complexity meta-feature compared to optimizing hyperparameters like the min_samples_leaf in an RF model, we provide strong evidence that complexity meta-features exert a greater influence on bias and variance dynamics.

A key insight from this analysis is that class overlap—captured by the N1 meta-feature—has a strong causal impact on model bias and variance. Reducing class overlap consistently led to better separability between classes, which resulted in a simultaneous decrease in both bias and variance. This simultaneous improvement is significant because it challenges the common notion of a strict bias-variance tradeoff, where typically, reducing one comes at the expense of increasing the other. Our findings show that addressing class overlap can mitigate both issues concurrently, suggesting that data complexity has a more direct and profound influence on RF model performance than hyperparameter tuning.

Table 1: Search Space for Model Hyperparameters

| Model | Hyperparameter | Type | Range |
|---|---|---|---|
| Random Forest | bootstrap | boolean | {true, false} |
| | max_features | float | [0.1, 0.9] |
| | min_samples_leaf | integer | [1, 20] |
| Support Vector Machine | kernel | nominal | {rbf, sigmoid} |
| | C | float | [$2^{-5}$, $2^{15}$] (log-scale) |
| | gamma | float | [$2^{-15}$, $2^{3}$] (log-scale) |
| Decision Tree | min_samples_split | integer | [2, 20] |
| | max_features | float | [0.1, 0.9] |
| | min_samples_leaf | integer | [1, 20] |
| Adaptive Boosting | algorithm | boolean | {SAMME, SAMME.R} |
| | learning_rate | float | [0.01, 2.0] (log-scale) |
| | n_estimator | integer | [50, 500] |
| Multi-Layer Perceptron | solver | categorical | {adam, lbfgs, sgd} |
| | batch$_s size$ | int | [1, 32, 64, 128] |
| | momentum | float | [0, 0.99] |
| | learning_rate_init | float | [1e-7, 0.5] |

Table 2: Summary of Linear Regression Analysis for RF Bias

| Feature | Coefficient | Standard Error | t-Statistic | P-Value | 95% CI |
|---|---|---|---|---|---|
| Intercept | 0.2183 | 0.000 | 891.237 | 0.000 | [0.218, 0.219] |
| N1 | 0.0945 | 0.000 | 191.829 | 0.000 | [0.094, 0.095] |
| L2 | -0.0144 | 0.000 | -39.106 | 0.000 | [-0.015, -0.014] |
| C2 | 0.0022 | 0.000 | 6.126 | 0.000 | [0.002, 0.003] |
| T2 | 0.0184 | 0.000 | 45.298 | 0.000 | [0.018, 0.019] |
| F1v | 0.0408 | 0.000 | 103.122 | 0.000 | [0.040, 0.042] |
| Density | 0.0053 | 0.001 | 10.479 | 0.000 | [0.004, 0.006] |
| min_samples_leaf | 0.0085 | 0.000 | 34.803 | 0.000 | [0.008, 0.009] |
| max_features | -0.0062 | 0.000 | -25.122 | 0.000 | [-0.007, -0.006] |
| bootstrap | 0.0033 | 0.000 | 13.470 | 0.000 | [0.003, 0.004] |

Table 3: Summary of Linear Regression Analysis for SVM Bias

| Feature | Coefficient | Standard Error | t-Statistic | P-Value | 95% CI |
|---|---|---|---|---|---|
| Intercept | 0.3651 | 0.000 | 1858.846 | 0.000 | [0.365, 0.365] |
| N1 | 0.0155 | 0.000 | 39.441 | 0.000 | [0.015, 0.016] |
| L2 | -0.0147 | 0.000 | -49.811 | 0.000 | [-0.015, -0.014] |
| C2 | -0.1139 | 0.000 | -393.628 | 0.000 | [-0.114, -0.113] |
| T2 | -0.0150 | 0.000 | -46.357 | 0.000 | [-0.016, -0.014] |
| F1v | 0.0211 | 0.000 | 66.800 | 0.000 | [0.021, 0.022] |
| Density | 0.0017 | 0.000 | 4.238 | 0.000 | [0.001, 0.003] |
| kernel | 0.0010 | 0.000 | 5.082 | 0.000 | [0.001, 0.001] |
| gamma | -0.0002 | 0.000 | -0.834 | 0.405 | [-0.001, 0.000] |
| C | -0.0011 | 0.000 | -5.822 | 0.000 | [-0.002, -0.001] |

Table 4: Summary of Linear Regression Analysis for DT Bias

| Feature | Coefficient | Standard Error | t-Statistic | P-Value | 95% CI |
|---|---|---|---|---|---|
| Intercept | 0.2317 | 0.000 | 884.947 | 0.000 | [0.231, 0.232] |
| N1 | 0.1054 | 0.001 | 210.777 | 0.000 | [0.104, 0.106] |
| L2 | 0.0008 | 0.001 | 1.035 | 0.301 | [-0.001, 0.002] |
| C2 | -0.0097 | 0.000 | -23.196 | 0.000 | [-0.011, -0.009] |
| T2 | 0.0096 | 0.000 | 33.231 | 0.000 | [0.009, 0.010] |
| F1v | 0.0399 | 0.001 | 62.465 | 0.000 | [0.039, 0.041] |
| Density | -0.0089 | 0.001 | -15.326 | 0.000 | [-0.010, -0.008] |
| min_samples_leaf | 0.0096 | 0.000 | 36.381 | 0.000 | [0.009, 0.010] |
| max_features | -0.0140 | 0.000 | -53.382 | 0.000 | [-0.015, -0.013] |
| min_samples_split | 0.0006 | 0.000 | 2.111 | 0.035 | [3.96e-05, 0.001] |

Table 5: Summary of Linear Regression Analysis for AB Bias

| Feature | Coefficient | Standard Error | t-Statistic | P-Value | 95% CI |
|---|---|---|---|---|---|
| Intercept | 0.2482 | 0.000 | 858.820 | 0.000 | [0.248, 0.249] |
| N1 | 0.0792 | 0.001 | 145.019 | 0.000 | [0.078, 0.080] |
| L2 | 0.0202 | 0.001 | 24.559 | 0.000 | [0.019, 0.022] |
| C2 | -0.0068 | 0.000 | -14.714 | 0.000 | [-0.008, -0.006] |
| T2 | 0.0070 | 0.000 | 21.935 | 0.000 | [0.006, 0.008] |
| F1v | 0.0348 | 0.001 | 49.584 | 0.000 | [0.033, 0.036] |
| Density | 0.0127 | 0.001 | 20.124 | 0.000 | [0.011, 0.014] |
| algorithm | 0.0001 | 0.000 | 0.369 | 0.712 | [-0.000, 0.001] |
| learning_rate | 0.0298 | 0.000 | 102.618 | 0.000 | [0.029, 0.030] |
| n_estimators | -0.0003 | 0.000 | -0.932 | 0.351 | [-0.001, 0.000] |

Table 6: Summary of Linear Regression Analysis for MLP Bias

| Feature | Coefficient | Standard Error | t-Statistic | P-Value | 95% CI |
|---|---|---|---|---|---|
| Intercept | 0.3016 | 0.000 | 630.484 | 0.000 | [0.301, 0.303] |
| N1 | 0.0896 | 0.001 | 98.011 | 0.000 | [0.088, 0.091] |
| L2 | 0.0112 | 0.001 | 8.203 | 0.000 | [0.009, 0.014] |
| C2 | -0.0231 | 0.001 | -30.203 | 0.000 | [-0.025, -0.022] |
| T2 | 0.0075 | 0.001 | 14.181 | 0.000 | [0.006, 0.009] |
| F1v | 0.0058 | 0.001 | 4.953 | 0.000 | [0.003, 0.008] |
| Density | 0.0035 | 0.001 | 3.271 | 0.001 | [0.001, 0.006] |
| batch_size | 0.0022 | 0.000 | 4.593 | 0.000 | [0.001, 0.003] |
| learning_rate_init | -0.0181 | 0.000 | -37.866 | 0.000 | [-0.019 , -0.017] |
| momentum | -0.0127 | 0.000 | -26.413 | 0.000 | [-0.014, 0.012] |
| solver | 0.0063 | 0.000 | 13.205 | 0.000 | [0.005, 0.007] |

Table 7: Summary of Linear Regression Analysis for RF Variance

| Feature | Coefficient | Standard Error | t-Statistic | P-Value | 95% CI |
|---|---|---|---|---|---|
| Intercept | 0.1131 | 0.000 | 750.759 | 0.000 | [0.113, 0.113] |
| N1 | 0.0485 | 0.000 | 160.217 | 0.000 | [0.048, 0.049] |
| L2 | -0.0127 | 0.000 | -55.915 | 0.000 | [-0.013 , -0.012] |
| C2 | -0.0162 | 0.000 | -73.166 | 0.000 | [-0.017, -0.016] |
| T2 | 0.0068 | 0.000 | 27.340 | 0.000 | [0.006 , 0.007] |
| F1v | 0.0251 | 0.000 | 103.171 | 0.000 | [0.025, 0.026] |
| Density | 0.0007 | 0.000 | 2.174 | 0.030 | [6.68e-05 , 0.001] |
| min_samples_leaf | -0.0012 | 0.000 | -7.722 | 0.000 | [-0.001 , -0.001] |
| max_features | 0.0067 | 0.000 | 44.291 | 0.000 | [0.006 , 0.007] |
| bootstrap | -0.0085 | 0.000 | -56.391 | 0.000 | [-0.009 , -0.008] |

Table 8: Summary of Linear Regression Analysis for SVM Variance

| Feature | Coefficient | Standard Error | t-Statistic | P-Value | 95% CI |
|---|---|---|---|---|---|
| Intercept | 0.1268 | 0.000 | 272.039 | 0.000 | [0.126 , 0.128] |
| N1 | 0.0174 | 0.001 | 18.591 | 0.000 | [0.016 , 0.019] |
| L2 | 0.0040 | 0.001 | 5.689 | 0.000 | [0.003 , 0.005] |
| C2 | -0.0738 | 0.001 | -107.462 | 0.000 | [-0.075 , -0.072] |
| T2 | 0.0205 | 0.001 | 26.726 | 0.000 | [0.019 , 0.022] |
| F1v | 0.0143 | 0.001 | 19.092 | 0.000 | [0.013 , 0.016] |
| Density | -0.0176 | 0.001 | -18.121 | 0.000 | [-0.020 , -0.016] |
| kernel | 0.0022 | 0.000 | 4.780 | 0.000 | [0.001 , 0.003] |
| gamma | 0.0013 | 0.000 | 2.880 | 0.004 | [0.000 , 0.002] |
| C | -0.0004 | 0.000 | -0.915 | 0.360 | [-0.001 , 0.000 ] |

Table 9: Summary of Linear Regression Analysis for DT Variance

| Feature | Coefficient | Standard Error | t-Statistic | P-Value | 95% CI |
|---|---|---|---|---|---|
| Intercept | 0.1955 | 0.000 | 1048.223 | 0.000 | [0.195 , 0.196] |
| N1 | 0.0825 | 0.000 | 231.488 | 0.000 | [0.082 , 0.083] |
| L2 | 0.0035 | 0.001 | 6.527 | 0.000 | [0.002 , 0.005] |
| C2 | -0.0256 | 0.000 | -85.813 | 0.000 | [-0.026 , -0.025] |
| T2 | 0.0035 | 0.000 | 17.174 | 0.000 | [0.003 , 0.004] |
| F1v | 0.0174 | 0.000 | 38.159 | 0.000 | [0.016 , 0.018] |
| Density | -0.0066 | 0.000 | -15.974 | 0.000 | [-0.007 , -0.006] |
| min_samples_leaf | -0.0060 | 0.000 | -32.334 | 0.000 | [-0.006 , -0.006] |
| max_features | -0.0090 | 0.000 | -48.385 | 0.000 | [-0.009 , -0.009] |
| min_samples_split | -0.0002 | 0.000 | -0.884 | 0.377 | [-0.001 , 0.000] |

Table 10: Summary of Linear Regression Analysis for AB Variance

| Feature | Coefficient | Standard Error | t-Statistic | P-Value | 95% CI |
|---|---|---|---|---|---|
| Intercept | 0.1603 | 0.000 | 594.868 | 0.000 | [0.160 , 0.161 |
| N1 | 0.0732 | 0.001 | 143.624 | 0.000 | [0.072 , 0.074] |
| L2 | 0.0092 | 0.001 | 12.006 | 0.000 | [0.008 , 0.011] |
| C2 | -0.0245 | 0.000 | -56.987 | 0.000 | [-0.025 , -0.024] |
| T2 | 0.0030 | 0.000 | 10.228 | 0.000 | [0.002 , 0.004] |
| F1v | 0.0128 | 0.001 | 19.618 | 0.000 | [0.012 , 0.014] |
| Density | -0.0106 | 0.001 | -17.949 | 0.000 | [-0.012 , -0.009] |
| algorithm | -0.0043 | 0.000 | -15.824 | 0.000 | [-0.005 , -0.004] |
| learning_rate | 0.0211 | 0.000 | 77.943 | 0.000 | [0.021 , 0.022] |
| n_estimators | -0.0070 | 0.000 | -25.910 | 0.000 | [-0.008 , -0.006] |

Table 11: Summary of Linear Regression Analysis for MLP Variance

| Feature | Coefficient | Standard Error | t-Statistic | P-Value | 95% CI |
|---|---|---|---|---|---|
| Intercept | 0.1149 | 0.000 | 366.086 | 0.000 | [0.114 , 0.116] |
| N1 | 0.0443 | 0.001 | 73.823 | 0.000 | [0.043 , 0.045] |
| L2 | 0.0187 | 0.001 | 20.762 | 0.000 | [0.017 , 0.020] |
| C2 | -0.0212 | 0.001 | -42.179 | 0.000 | [-0.022 , -0.020] |
| T2 | 0.0023 | 0.000 | 6.709 | 0.000 | [0.002 , 0.003] |
| F1v | -0.0001 | 0.001 | -0.147 | 0.883 | [-0.002 , 0.001] |
| Density | -0.0104 | 0.001 | -15.036 | 0.000 | [-0.012 , -0.009] |
| batch_size | -0.0010 | 0.000 | -3.303 | 0.001 | [-0.002 , -0.000] |
| learning_rate_init | 0.0210 | 0.000 | 66.692 | 0.000 | [0.020 , 0.022] |
| momentum | 0.0051 | 0.000 | 16.160 | 0.000 | [0.004 , 0.006] |
| solver | -0.0097 | 0.000 | -30.669 | 0.000 | [-0.010 , -0.009] |

