# OpenReview forum: "Attributing Model Behavior: The Predominant Influence of Dataset Complexity Over Hyperparameters in Classification"
_ICLR.cc/2025/Conference — Submitted to ICLR 2025_

### Official Review · Reviewer_PZm3 · 2024-10-28

**Soundness:** 2
**Presentation:** 4
**Contribution:** 2
**Rating:** 3
**Confidence:** 5

**Summary:**

The paper focuses on the attribution of predictive model behavior. In particular, it describes a comprehensive empirical comparison of the influence of hyperparameters and dataset meta-features on the bias and variance of classifiers. The analysis utilizes functional ANOVA. The main result is that most of bias and variance can be attributed to dataset characteristics, as opposed to hyperparameters.

**Strengths:**

- Well structured and clearly written paper.
- A many ways a very comprehensive experiment.
- Tackles an important issue for ML practitioners.

**Weaknesses:**

I have two main concerns. The first is that the experiments, while in some ways very comprehensive, are in other ways very limited:

- Only two classification algorithms.
- No missing values in the dataset is a very strong criterion.
- A limited number of complexity measures.

The second is that when carefully interpreted, the results are not that general or actionable:
- N1 is in essence a classifier (probably along the lines of LDA). So the results can basically be summarized that most of the bias and variability of RF and SVM can be explained by running another reasonable classifier and seeing how it performs. That of course makes perfect sense, but it can also be derived from what we already know, that classifiers tend to perform similarly (the differences between classifiers are less than the differences between datasets).
- We should be more careful when interpreting the result that model performance can be attributed more to dataset characteristics than to hyperparameters. First, it is the nature of commonly used classifiers that they are relatively robust in terms of hyperparameter selection - being easy to tune is what makes them popular. Second, . And third, the range of several parameters is limited. For example, would results change if max_features was allowed to go below 0.1 or above 0.9? Or if 20 different kernels were considered? Similarly, the experiments are limited to 1500 features, which diminishes the importance of regularization.
- In practice, I can in most cases freely tune the parameters and select models. I can't really change my problem (or dataset) though.
- The paper does not consider model selection, which I would in this context consider as part of hyperparameter tuning. I would not be surprised that a lot more can be attributed to model selection than to tuning the parameters in this paper. Choosing a different model is also actionable.

There are also other methodological concerns (see Questions).

**Questions:**

1. fANOVA is one of many ways of decomposing model predictions. Why this approach? And are there any potential issues due to taking into account pairwise interactions only?
2. l. 60: How can normalization or scaling affect the intrinsic complexity of the datasets? Intuitively, wouldn't a reasonable measure of complexity be invariant to these? Of the three in the paper, C1 is invariant. T2 is not, because it depends on PCA, but that just makes me wonder if T2 even makes sense. I wouldn't want my dataset to become more complex, just because I convert a feature from meters to centimeters. N1 is based on a two-sample test, so, while I didn't go into the details of the test, I'd assume that we'd like all of our two sample tests also to be invariant to scaling. ... To clarify, I'm not criticizing the choice of not scaling, we can argue either way. But I am concerned about the use of complexity metrics that are not invariant.
4.But not normalizing does have an effect on SVM and regularization? This is not how we would apply SVM in practice.
5. N1 is a bit outdated. Two sample tests have progressed a lot in recent years. In particular, tests based on machine learning models directly or using classification performance as a proxy.
6. It seems that N1 would fail to be attributed when the dataset is so complicated that RF and SVM can perform well, but the test in N1 doesn't?
7. The attribution to C1 for SVM is to me the most surprising result in the paper. Any explanations of this difference between SVM and RF? How correlated are N1 and C1?
8. l. 658: How reasonable is this assumption that there are no hidden confounding factors?
9. Parameter configurations were generated by sampling uniformly and independently from each hyperparameter range? I'm asking because of the following scenario. Let's say that the optimal range for a parameter is relatively narrow (0 - 0.05), while after that (0.05 - 1.0) the model performs majority class baseline poorly. Because the range of good values is small, this, as a variable in a regression, would not be that important. So, it would not get much of an attribution in the experiment, but it is definitely important in practice. In other words, isn't the importance of a hyperparameter determined by the difference it can make, not by the variability of the performance over some arbitrary set of its values?

---

> ### Author Response · Authors · 2024-11-26
>
> Dear Reviewer, thank you for your detailed feedback and thoughtful suggestions on our manuscript. We value your insights, which have significantly contributed to refining our work. Below, we address your specific concerns:
>
> Experimental Scope:
> 1. Limited Number of Classification Algorithms: We acknowledge the limitation in the number of algorithms initially analyzed. In the revised manuscript, we have addressed this by extending our analysis to include three additional classifiers: Multi-Layer Perceptron (MLP), Decision Tree (DT), and AdaBoost (AB). This broader scope strengthens the comprehensiveness and generalizability of our conclusions.
>
> 2. Criterion of No Missing Values: While we recognize that the criterion of excluding datasets with missing values may appear restrictive, it was a deliberate decision to avoid potential alterations to dataset complexity from imputation or other preprocessing methods. Such alterations could confound our analysis of intrinsic dataset characteristics.
>
> 3. Limited Complexity Measures: We appreciate your observation regarding the limited number of complexity measures in our initial study. Complexity meta-features in literature are categorized into six groups: Feature-based measures, Linearity measures, Neighborhood measures, Network measures, Dimensionality measures, and Class imbalance measures. Initially, we covered only three categories. In the revised version, we extend our analysis to incorporate measures from all six categories, thus addressing this limitation comprehensively.
>
> Generalizability and Interpretation of Results
> 1. Misconception Regarding N1: We respectfully disagree with the interpretation that N1 is "in essence a classifier". N1 is not a classifier but rather a meta-feature that quantifies dataset complexity by assessing class separability in the feature space. Specifically, N1 calculates the "Fraction of Borderline Points" by constructing a Minimum Spanning Tree (MST) and evaluating how many points are connected to others of different classes. While its goal aligns with classifiers like LDA (in assessing separability), its method is geometric, not predictive. We have revised the manuscript to ensure this distinction is clear.
>
> 2. Robustness of Classifiers and Dataset Characteristics: We agree that popular classifiers are designed to be robust to hyperparameter choices, which partially explains their broad adoption. However, our findings emphasize that dataset characteristics (e.g., class overlap, dimensionality) fundamentally constrain performance, regardless of hyperparameter tuning. Hyperparameter tuning can optimize performance within the limits set by dataset complexity, but it cannot overcome intrinsic challenges such as high class overlap or imbalance. We do not aim to dismiss the importance of hyperparameter tuning but to emphasize that intrinsic dataset properties often have a greater impact.
>
> 3. Addressing Dataset Complexity: The assertion that dataset complexity cannot be changed is not entirely accurate. Characteristics such as class overlap and class imbalance can be addressed using advanced sampling techniques like Tomek Links or SMOTE variants, which modify the dataset to enhance separability and balance. Similarly, dimensionality-related complexity can be mitigated through methods such as PCA, t-SNE, or feature selection, which reduce feature space while retaining critical information.  Assuming it is indeed impossible to alter dataset complexity, this then emphasizes the importance of further research in this area to explore the untapped potential of data-centric strategies for improving model performance. Our study seeks to highlight these opportunities and encourage further investigation into leveraging dataset modifications for better outcomes.
>
> 4. Model Selection vs. Tuning: We agree that the choice of model is an actionable strategy, but the primary focus of our study was to compare the relative influence of dataset characteristics and hyperparameters on model behavior, independent of model selection. This remains an important contribution, as understanding the influence of data properties can help practitioners make informed decisions when choosing models and tuning parameters.

---

> ### Author Response · Authors · 2024-11-27
>
> Methodological Concerns
>
> 1. Choice of fANOVA: We selected fANOVA due to its widespread use in the literature and its suitability for quantifying hyperparameter importance in high-dimensional parameter spaces. However, we acknowledge that higher-order interactions may remain unexplored and as such we have explicitly highlighted this in the revised manuscript, proposing that future work should explore this further.
>
> 2. Effect of Normalization on Complexity Measures: We decided not to normalize our data because normalization can indeed influence certain complexity meta-features of a dataset by altering its geometry, distances, or statistical properties. For instance, normalization changes the distances between data points, which can subsequently affect the structure of the minimum spanning tree used to compute measures like N1. Similarly, linearity-based complexity measures, which assess how well a dataset can be separated by linear classifiers, are impacted by normalization because the relative scaling of features influences the linear separability of the classes.
>
> 3. Relevance of N1: We respectfully disagree with the characterization of N1 as outdated and its comparison with Two sample tests. N1 is not directly related to two-sample tests, nor are two-sample tests an advancement of it. They serve distinct purposes: two-sample tests, such as the Kolmogorov-Smirnov (KS) test or Hotelling's T² test, are designed to compare the statistical properties (e.g., means, variances, cumulative distributions) of two distributions to determine if they are significantly different. In contrast, N1 evaluates the spatial arrangement of data points within a multi-dimensional feature space, focusing on the proportion of points that are close to the decision boundary between classes. This measure does not involve comparing statistical distributions but rather assesses the inherent separability of the data.
>
> 4. Attribution of Class Imbalance to SVM: We were not surprised by the attribution of complexity measures (like class imbalance) to SVM, as it is well-known that SVMs are particularly sensitive to class imbalance, while Random Forests are more robust but not entirely immune. We have referenced existing studies in the revised paper to support this attribution.
>
> 5. Correlation Between Class Imbalance and Overlap: Class imbalance and class overlap are not inherently correlated, but they can coexist and amplify each other.
>
> 6. Assumption of No Hidden Confounding Factors: Our experiments were conducted in a controlled computational setting, where key sources of variability were explicitly measured or fixed. This reduces the likelihood of significant unobserved confounders and as such we consider our assumption of causal sufficiency reasonable. Nonetheless, we acknowledge this assumption as a possible limitation.
>
> 7. Uniform Sampling of Hyperparameters: While our approach provides broad coverage of the hyperparameter space, we acknowledge that uniform sampling may not fully capture the contribution of hyperparameters with narrow optimal ranges. This is a valid concern, and in the revised manuscript, we have made a note about the potential limitations of this approach. In future work, we plan to explore more sophisticated sampling strategies, such as adaptive sampling, to better capture the impact of hyperparameters with narrow but crucial effective ranges.
>
> We hope the revisions and clarifications effectively address your concerns and enhance the value of our work. We believe the improvements strengthen our findings and warrant consideration for acceptance.

---

> ### Author Response · Authors · 2024-11-27
>
> To further address your comment on how to address data complexity,  we have expanded the result summary section in the revised version of our paper, referencing a recent study that investigated optimal resampling techniques for imbalanced and complex datasets. Their findings show that for datasets with high class overlap (high N1) and imbalance, applying methods like SMOTE-TomekLinks or SMOTE-PSO improved the performance of tree-based models such as Random Forest and Decision Trees. However, the authors of that study could not explain the underlying reasons for these improvements.
>
> Our findings provide the missing explanation, revealing that the N1 complexity metric, which measures class overlap, plays a dominant role in influencing bias and variance, particularly in tree-based models. This insight not only supports their findings but also provides practitioners with valuable guidance on how other complexity meta-features and model hyperparameters impact model performance.
>
> We hope this clarification addresses your concern and enhances the practical relevance of our work.

---

### Official Review · Reviewer_maFk · 2024-10-30

**Soundness:** 1
**Presentation:** 2
**Contribution:** 2
**Rating:** 3
**Confidence:** 4

**Summary:**

Authors compare impact of dataset complexity and hyperparameter tuning on the performance of binary Random Forest and SVM classifiers. They perform extensive experiments which support their finding that the dataset complexity has dominant impact on the performance.

**Strengths:**

- extensive experiments

- clearly written, easy to read

- the topic of bias/variance tradeoff is very important

**Weaknesses:**

I think the paper sends, in essence, a wrong message to readers.
Authors are basically suggesting that hyperparameter optimization
isn't useful, which I disagree with.  As I write this, I am tuning
hyperparameters of a neural network classifier, and the AUROC has gone
from 0.55 to 0.75, exclusively due to the (gradually improving) choice
of hyperparameters. The conclusion is at best narrowly limited to SVM
and RF, but that makes the manuscript 1) not that useful, given
limited scope 2) misleading, since many readers may walk away with a
wrong impression that the conclusions apply generally

To be clear, not disagreeing with the idea that hyperparameter tuning
has a natural limit, and going beyond that may require additional or
different data. But the paper leaves an impression to the reader that
hyperparameter tuning doesn't help in general, which I disagree with. At a minimum, the title should say that the results are limited to SVM and RF binary classifiers.

Also keep in mind that "optimizing dataset complexity" is a vague and
hardly actionable advice. I personally don't quite know how to
optimize dataset complexity, whereas hyperparameter tuning is well
understood. This should be clearly stated/discussed.

**Questions:**

No questions, but recommendations for future submission:

- drop the analysis of dataset complexity vs. hyperparameters; just
  focus on the impact of hyperparameters on the bias/variance
  trade-off. It is sufficiently important topic on its' own. Impact of dataset complexity is not useful because 1) it is well understood that dataset complexity has major impact on classifier performance 2) it is not clear what to do about it.

- include multi-class problems

- include XGBoost and neural networks in the analyses. I don't think
  one can make practically useful conclusions without including those
  state-of-the-art classifiers. I would personally also add logistic
  regression

- consider adding image classification datasets in your analyses. If
  the behavior - in terms of bias/variance, and hyperparameter tuning
  contribution - is quite different from tabular data, that would be a
  valuable result

I think this *could* become a good paper, but not without extensive
revision, which is not feasible in the ICLR timeframe.


Minor points

3.3
---

- 2 out of 3 is not really "generally". Please just be specific: for
  N1 and T1, higher values indicate greater classification
  difficulty. For C1, lower values indicate greater classification
  difficulty. That would be simpler and easier to read.

  "Higher values of these meta-features generally indicate greater
  classification difficulty (except for C1)."

4.2.2
-----

- "When considering variance, a similar trend emerges. According to
  the fANOVA results (Figure 3b), C1 continues to dominate, accounting
  for 37.78% of the variability in variance."

  I wouldn't call this a similar trend. For bias, C1 accounts for 71%,
  for variance, 38%. Please point out that the C1 impact on variance is significantly lower than on bias. This suggests other factors play greater role. Discuss what those factors might be.

---

> ### Author Response · Authors · 2024-11-26
>
> Dear Reviewer, thank you for your detailed and constructive feedback on our work. We appreciate the time and effort you put into reviewing our manuscript and have made significant revisions to address the points raised. Below, we provide a detailed response to your concerns.
>
> Scope and Interpretation of Results
> 1. On the Role of Hyperparameter Optimization: we respectfully disagree with the assertion that our paper suggests hyperparameter optimization is unimportant or ineffective. Rather, the goal of our study is to highlight that, relative to intrinsic data complexity, the contribution of hyperparameter tuning to bias-variance dynamics is smaller but still statistically significant. Our findings are not meant to downplay the importance of hyperparameter optimization but rather to emphasize the often-overlooked influence of dataset characteristics, such as class overlap, on model performance.
>
> To ensure this distinction is clear, we have revised the manuscript to explicitly state that hyperparameter tuning remains a critical step in optimizing model performance but may yield diminishing returns in the presence of complex or poorly structured datasets. This perspective aligns with your observation that hyperparameter tuning can substantially improve performance, as in your example with AUROC improvements. Our conclusions advocate for a more balanced approach that considers both hyperparameter tuning and addressing dataset complexities.
>
> 2. Generality of the Results: We acknowledge the limited scope of the initial submission and appreciate your feedback on expanding the analysis. In the revised version of the paper, we have incorporated additional experiments with a Multi-layer Perceptron (MLP), Decision Tree (DT), and AdaBoost, covering a broader range of classification algorithms. Across these models, our findings remain consistent: complexity meta-features of datasets, particularly class overlap, exert a dominant influence over hyperparameters in determining bias and variance.
>
> We also acknowledge your suggestion to consider multiclass and image classification datasets. While these are beyond the scope of our current study, they represent valuable directions for future work.
>
> Actionability of Dataset Complexity Insights
> 1. Practicality of Addressing Dataset Complexity:  Contrary to the assertion that "optimizing dataset complexity" is vague or impractical, our study aims to draw attention to a growing body of research proposing concrete methods for mitigating intrinsic data complexities. For example: Class overlap and imbalance can be addressed using sampling approaches, such as Tomek Links or SMOTE variants. Also, dimensionality can be reduced through techniques like PCA, t-SNE, or feature selection methods.
>
> Our work is positioned as complementary to these efforts, emphasizing the importance of understanding and addressing complexity as part of the broader data preparation pipeline.
>
> Focus of the Paper
> 1. Importance of Complexity Analysis: While it is intuitive that data complexity significantly impacts classifier performance, we argue that empirical and systematic studies quantifying this relationship remain sparse. Our work contributes to filling this gap by rigorously comparing the relative influences of data complexity and hyperparameters across diverse datasets and algorithms.
>
> Recent initiatives, such as the establishment of the Data-centric Machine Learning Research journal by the Journal of Machine Learning Research (JMLR), highlight the renewed focus on understanding the interplay between data properties and model performance. Our findings are consistent with this growing emphasis and provide empirical evidence to support the prioritization of data-centric approaches in certain scenarios.
>
> We acknowledge your suggestion to focus exclusively on hyperparameter analysis. However, we believe that understanding the interplay between dataset complexity and hyperparameters offers a more holistic perspective on model optimization, which is valuable for both researchers and practitioners.
>
> Conclusion
> We believe that the revisions and clarifications provided in the updated manuscript address the concerns raised in your review. We hope that these updates improve the clarity, scope, and practical relevance of our work. Thank you again for your valuable feedback, which has greatly contributed to enhancing the quality of our study.

---

> ### Author Response · Authors · 2024-11-27
>
> To further address your comment on how to address data complexity, we have expanded the result summary section in the revised version of our paper, referencing a recent study that investigated optimal resampling techniques for imbalanced and complex datasets. Their findings show that for datasets with high class overlap (high N1) and imbalance, applying methods like SMOTE-TomekLinks or SMOTE-PSO improved the performance of tree-based models such as Random Forest and Decision Trees. However, the authors of that study could not explain the underlying reasons for these improvements.
>
> Our findings provide the missing explanation, revealing that the N1 complexity metric, which measures class overlap, plays a dominant role in influencing bias and variance, particularly in tree-based models. This insight not only supports their findings but also provides practitioners with valuable guidance on how other complexity meta-features and model hyperparameters impact model performance.
>
> We hope this clarification addresses your concern and enhances the practical relevance of our work.

---

### Official Review · Reviewer_vQV7 · 2024-11-04

**Soundness:** 3
**Presentation:** 4
**Contribution:** 2
**Rating:** 5
**Confidence:** 3

**Summary:**

This paper explores the relative influence of dataset complexity and hyperparameters on classification model behavior, specifically for RF and Kernel SVM. \
Authors utilize the fANOVA framework and OLS to quantify the influence on bias and variance brought by dataset complexity and hyperparameters. \
Based on the analysis across 290 datasets and 304 hyperparameter configurations, the study finds that dataset complexity meta-features—such as class overlap, data sparsity, and class imbalance—have a more substantial impact on bias and variance than hyperparameters.

**Strengths:**

The paper addresses a gap by directly comparing the impacts of dataset complexity and hyperparameters on model performance. Previous studies have often examined these factors separately, but this research provides a unified framework to assess their relative influence on bias and variance.

The paper comprehensively considered nearly 300 datasets and over 300 configurations, enabling a more convincing conclusion.

Apart from the numerical results, the paper also includes very detailed arguments of why this happens and what this indicates.

Some of the estimated coefficients shown in the OLS summary table do align with our common understanding of how random forests deal with bias-variance tradeoffs, e.g. coefficients associated with min samples leaf, bootstrap, and max features.

**Weaknesses:**

1. lack of details on the experiment design.

 (1) as the paper claims, nearly 300 datasets of varying sample sizes, response categories, and feature dimensions are used. Why are they comparable? I believe 0-1 loss is not a typical loss function people use for multiclassification problems.  And high-dimensional datasets wouldn't react necessarily the same as n>p datasets in terms of hyperparameters.

(2) hyperparameters like C and gamma have huge variations in scale. How is it being included in OLS? Is it logarithmized?

 (3) For readers not familiar with meta-features of datasets, it would be very helpful to at least sketch some general ideas of how these meta-features are defined. Are those features immune to data transformation?  The same for how fANOVA works.


2. lack of details on why the experiments are conducted in such a way.

 (1) In my perspective, the number of trees is RF's one of the most important hyperparameters. Why is this not considered?

(2) I believe the neural net is the framework that people are most curious about. The authors also mention it in the introduction. Why is that not considered?

(3) Based on the pymfe package, there are plenty of meta-features that characterize data complexity from different perspectives. Why specifically these 3, N1, T2, C1, are chosen?

3. Some of the results that are confusing to me.

(1) if those meta-features are immune to data transformation, how can we benefit from your research even though we know that data complexity itself is much more important than tuning hyperparameters? if not, shouldn't you include some examples of how bias and variance are reduced after some preprocessing of the data that reduces data complexity? For example, class imbalance issues can be alleviated by reweighing samples or bootstrapping.

In general, I do agree that the data's quality is much more important than tuning parameters. If the data is always linearly separable, I believe logistic regression would suffice. It's just the data quality is not something we can work on but the model and the model's parameter choice. Please correct me if I am wrong.

 (2) Based on your OLS example, the features included are all significant neq to 0. If the trend is determined, does it mean that choosing a smaller c or some certain kernel can always help with the prediction?

**Questions:**

please check the weakness.

---

> ### Author Response · Authors · 2024-11-25
>
> Dear Reviewer, Thank you for taking the time to review our work and for providing detailed and thoughtful feedback. Below, we address your concerns regarding the experimental design, choice of methodology, and interpretation of results.
>
> Lack of Details on the Experimental Design
>
> 1. Comparability of Datasets and Use of 0-1 Loss: The use of nearly 300 datasets with varying sample sizes, response categories, and feature dimensions was a deliberate choice aimed at ensuring the generalizability of our findings across a broad spectrum of real-world scenarios. To assess whether these datasets were comparable, we conducted an initial experiment using fANOVA to measure the influence of factors such as sample size and feature dimensions on model performance. The results indicated that while these factors have some influence, their effects were overshadowed by the impact of intrinsic complexity meta-features. This justified our decision to treat the datasets as comparable despite differences in their characteristics.
>
> Regarding the use of 0-1 loss: although it is not a typical choice for multiclass classification, it is a standard metric used in studies that decompose bias and variance in classification problems. Relevant citations supporting this choice are provided in the background section of our paper.
>
> 2. Inclusion of Variables in OLS: to ensure the comparability of independent variables with differing scales, we standardized all variables before including them in the OLS analysis. This was mentioned in the methodology section of our paper.
>
> 3. Explanation of Meta-Features and fANOVA: We have revised the manuscript to include additional descriptions of how complexity meta-features are computed. While some meta-features (e.g., class separability) remain unaffected by certain transformations, others others can change significantly depending on the type of transformation applied. This sensitivity reinforces the importance of selecting appropriate preprocessing steps for specific complexity characteristics. For example operations like row shuffling  typically leave most global meta-features (e.g., number of features, number of samples) intact. However, techniques like PCA or t-SNE change the number of features and their correlations, significantly altering meta-features like dimensionality.
>
> Lack of Details on Experimental Design Choices
> 1. Exclusion of the Number of Trees in Random Forest (RF): the number of trees was not included as a hyperparameter in our analysis because we focused on the three most influential hyperparameters identified in prior research on RF. These hyperparameters have been widely documented as having the most significant impact on RF performance. References to this supporting literature are included in our introduction and methodology sections.
>
> 2. Neural Networks (NN) in the Scope of the Study: we acknowledge the importance of neural networks in the current machine learning landscape. While our original submission did not include results for neural networks due to time constraints, we have since incorporated analysis on Multi-layer Perceptrons (MLP) in the revised version of the paper. Results for Decision Trees (DT) and Adaptive Boosting (AB) have also been updated for completeness.
>
> 3. Choice of Complexity Meta-Features: our initial scope was limited to analyzing three commonly discussed complexity meta-features (N1, T2, and C1), based on their prevalence in the literature. However, we appreciate the reviewer’s suggestion to consider additional categories of complexity features. We have revised the paper to include complexity meta-features related to density, class separability, and feature overlap. This expanded analysis strengthens the breadth and depth of our findings as it covers the six main categories of complexity meta-features defined in the literature.
>
>
> Clarifications on Confusing Results
> 1. Applicability of Complexity Meta-Features and Data Transformations: Complexity meta-features are not immune to data transformations. Many preprocessing techniques can mitigate specific complexities and improve model performance. For example, sampling techniques such as TomeLinks can be used to mitigate the presence of class imbalance and overlapping in classification datasets. While our work focuses on quantifying the relative influence of intrinsic complexity and hyperparameters, we acknowledge the importance of preprocessing in addressing these complexities. The introduction section of our revised paper highlights relevant studies that explore the influence of preprocessing on complexity. Hence, we respectfully disagree with the assertion that data quality cannot be improved.
>
> 2. Hyperparameter Trends: our OLS results indicate that while hyperparameters have a statistically significant influence on bias-variance dynamics, their contribution is relatively small compared to the impact of dataset complexity. This does not imply that specific hyperparameter settings are universally optimal.

---

> > ### Comment · Reviewer_vQV7 · 2024-11-26
> >
> > Thank you for your reply.
> >
> > For my question 3(1):
> >
> > The authors replied "Many preprocessing techniques can mitigate specific complexities and improve model performance. " While I meant to say that the authors should provide some examples **based on the research results** in this paper to improve the model performance. For example, if higher N1 increases bias, can we actually do data transformations to reduce N1 and the model performance indeed improves?

---

> > > ### Author Response · Authors · 2024-11-27
> > >
> > > Dear Reviewer, thank you for your valuable feedback and for highlighting the need to connect our findings to actionable recommendations. We appreciate your suggestion to provide examples demonstrating how reducing specific complexities, such as N1, can improve model performance.
> > >
> > > While the primary scope of our study was to empirically quantify and compare the influence of data complexity features and hyperparameters on bias-variance dynamics, we acknowledge the practical value of such examples. To address your concern, we have expanded the result summary section in the revised version of our paper, referencing a recent study that investigated optimal resampling techniques for imbalanced and complex datasets. Their findings show that for datasets with high class overlap (high N1) and imbalance, applying methods like SMOTE-TomekLinks or SMOTE-PSO improved the performance of tree-based models such as Random Forest and Decision Trees. However, the authors of that study could not explain the underlying reasons for these improvements.
> > >
> > > Our findings provide the missing explanation, revealing that the N1 complexity metric, which measures class overlap, plays a dominant role in influencing bias and variance, particularly in tree-based models. This insight not only supports their findings but also provides practitioners with valuable guidance on how other complexity meta-features and model hyperparameters impact model performance.
> > >
> > > We hope this clarification addresses your concern and enhances the practical relevance of our work.

---

### Official Review · Reviewer_SZJd · 2024-11-05

**Soundness:** 1
**Presentation:** 2
**Contribution:** 1
**Rating:** 1
**Confidence:** 5

**Summary:**

The paper proposes an analysis of the influence of hyperparameter tuning and training "data complexity" (the author call it "complexity meta-features") on the performance of two classic classification algorithms:SVMs and Random forests. the paper includes  run extensive experiments on 290 OpenML tabular datasets. The author's end with a summary of their findings: dataset complexity matters the most.

**Strengths:**

1) the paper is easy to read
2) The paper confirms a fact that is well know by most data science / ML prectioners, the complexity of the data set matters for classification performance.

**Weaknesses:**

1) The content, experiments and conclusions of the papers are very outdated. It reads like a paper that was written 15-20 years ago. Most citations are from many years ago. Hence the contribution are practically irrelevant to the current state of ML / data science in 2024. Hence there is no significant contribution or relevance to the ICLR community.

2) Furthermore, the main conclusion of the paper is that dataset complexity (class overlap, dimensionality, etc) matters when training a classifier (RF or SVM). These are well known fact that are thought in introductory ML class and hence there is no new information provided here.

**Questions:**

I dont have any follow-up questions.

---

> ### Author Response · Authors · 2024-11-25
>
> Dear Reviewer, thank you for your thoughtful review and for sharing your concerns regarding the contributions and relevance of our paper. We appreciate the opportunity to clarify and address the points raised.
>
> On the Perceived Outdated Nature of the Research:
>
>  We respectfully disagree with the characterization of our work as outdated or irrelevant to the current state of machine learning and the ICLR community. While it is true that studies on the impact of intrinsic data characteristics have historically been less prominent, there has been a growing resurgence of interest in data-centric machine learning. Recent initiatives, such as the establishment of the Data-centric Machine Learning Research (DMLR) journal by the Journal of Machine Learning Research (JMLR), underscore the renewed emphasis on understanding how data quality and complexity influence model performance. Our work contributes to this emerging focus by empirically quantifying the comparative effects of data complexity and hyperparameters across a wide variety of classification algorithms (paper has been revised to include results for Decision Tree, AdaBoost and Multi-Layer Perceptron) and datasets. The methods and conclusions presented are not only timely but also highly relevant as they provide actionable insights into how data characteristics impact key components of model behavior, including bias and variance. These findings are especially pertinent for practitioners and researchers aiming to optimize model performance in increasingly complex and data-driven applications.
>
> On the Novelty of Findings Related to Data Complexity:
>
>  We agree that the importance of dataset complexity—such as class overlap, dimensionality, and feature interaction—is a widely acknowledged principle in the machine learning community. However, our primary contribution lies not in asserting this principle but in rigorously quantifying its impact relative to hyperparameter tuning. While many practitioners intuitively understand the significance of data complexity, there has been a lack of systematic empirical research measuring its effect sizes or disentangling its influence from other factors like hyperparameter configurations. Our study fills this gap by employing robust analytical techniques, such as functional analysis of variance (fANOVA) and Ordinary Least Squares (OLS) regression, to demonstrate that dataset complexity often outweighs hyperparameter optimization in its impact on classification performance. This empirical validation moves beyond intuition, providing a quantitative foundation for prioritizing efforts to mitigate data complexity issues over exhaustive hyperparameter tuning.
>
> Relevance to the ICLR Community:
>
>  Understanding the interplay between data characteristics and algorithm performance is crucial for advancing the field of machine learning. By framing our findings within the context of bias-variance and presenting evidence-based recommendations, our study aligns with the ICLR community's commitment to driving meaningful progress in both theoretical and applied machine learning. We believe our work addresses a critical, yet underexplored, aspect of machine learning research that complements current trends in algorithmic and systems-level innovation. As such deserves consideration.

---

### Meta-Review · Area_Chair_1r9Q · 2024-12-22

**Metareview:**

The paper analyzes in a large empirical studies the impact of dataset complexity and hyperparameter tuning on the performance of two classic learning algorithms, Random Forest and Support Vector Machines.

Unfortunately, the paper does not contribute anything new and the main conclusions from the experiments are rather well-known and well-acknowledged by the community.

**Additional Comments On Reviewer Discussion:**

The rebuttal from the Authors came very late, so the Reviewers had a limited chance to discuss the paper further with the Authors. During the reviewer discussion phase, the most critical Reviewers confirmed their initial doubts. The most positive Reviewer did not champion the paper.

---

### Decision · Program_Chairs · 2025-01-22

Reject